# Review on Mandibular Muscle Kinematics

**DOI:** 10.3390/s22155769

**Published:** 2022-08-02

**Authors:** Beatriz Martínez-Silva, Montserrat Diéguez-Pérez

**Affiliations:** 1Faculty of Biomedicine and Health Sciences, European University of Madrid, 28670 Villaviciosa de Odón, Spain; 21775473@live.uem.es; 2Preclinical Dentistry Department, Faculty of Biomedicine and Health Sciences, European University of Madrid, C. Tajo s/n, 28670 Villaviciosa de Odón, Spain

**Keywords:** motion capture system, mandible kinematics, oral function, dentistry, biomechanics, vicon system

## Abstract

The complexity of mandibular dynamics encourages constant research as a vehicle to improve oral health. The gold standard motion capture system might help us to understand its functioning and its relation to body position, aiming to perform an exhaustive bibliographic review in the Dentistry field. Six different electronic databases were used (Dentistry & Oral Sciences Source, Scopus, Web of Science, PubMed, CINAHL and SPORTDiscus) in April 2022. The selection criteria includes a biography, critical analysis, and the full text from 1984 to April 2022, based on the odontological gold standard, whether or not in combination with additional devices. Clinical cases, bibliographic reviews or meta-analysis and grey literature were excluded. The checklist of the critical assessment methodology by Joanna Brigs was used (JBI). After choosing scientific articles published in peer-reviewed journals, 23 out of 186 investigations were classified as eligible with a total of 384 participants. The issue being addressed is related to the speech properties, posture and body movement in relation to dento-oro-facial muscle and facial analysis, mandibular kinematics and mandibular dynamics during the mastication process. The markers arrangement depends on the dynamic to be analysed. From a physiologic and pathologic perspective, the applications of the optic system are relevant in Dentistry. The scarcity of literature obtained implies the need for future research.

## 1. Introduction

Biomechanics is the science that studies, analyses and interprets the activity performed by living things, in particular the human activity, aiming to collect data quantitatively on forces and the effects caused by its application over the musculoskeletal system during motor activity performance [1]. The research on the motor activity in human beings allows the system functional capacity that intervenes in the physical action to be understood in order to be able to categorize it as physiologic or pathologic, and to be capable of establishing and verifying biomedical therapeutic procedures [1,2,3]. In order to carry out this analysis, there are human motion capture systems that gather spatio-temporal information that is digitally represented and an optical system, which is the most appropriate for recording human activity [4]. The optical systems currently require the use of large laboratories equipped with a multicamera synchronized system consisting of two or more units located in different angles, from which the location and the space is calculated; the higher the number cameras used, the higher the measurement accuracy and reliability will be [5]. Within this motion capture technique based on images, we can include a technique based on the use of reflective passive markers of an infrared lamp that are located in three strategic anatomic landmarks where the three-dimensional location of these is calculated through triangulation [4,5].

The reference (gold standard) motion capture system is currently Vicon^®^, due to its high level of accuracy of generated data [6], with a location average error in the space under 0.5 mm [7]. This is an optoelectronic capture system, which uses infrared cameras and passive markers, which process the information through software that analyses the reproduction of the movement itself [8]. This technology finds application in the clinical analysis of the biomedical sciences [9,10,11,12,13,14,15], including Dentistry. 

The study of mandible dynamics, as well as the behaviour of every structure that makes up the stomatognathic system, has been the subject of study for its analysis, understanding and diagnosis of dento-oro-facial problems [16,17,18,19,20,21]. Oral health implies that all the components that make up this system and its relationship with the environment work in balance and harmony, however, the existence of an alteration in any of the elements can result in the development of a dento-oro-facial disorders [22,23]. A certain alteration trigger of the Stomatognathic Apparatus is the existence of an occlusal discrepancy between maximum intercuspation (MI) and the centric relation (CR), closely related to temporomandibular joints disorders [24,25,26]. Furthermore, the difficulties in muscle chains of the Stomatognathic system, intrinsically linked to oral parafunction like bruxism, are due to harmful muscle adaptive repositioning [23]. As a result of this situation, the presence of pathological mandibular eccentric movements is favourable [27].

There are multiple pieces of research that relate the motion of the lower jaw, occlusion, or stomatognathic system disorders to body posture [28,29,30,31]. The existence of occlusal alterations as the existence of variation in MI and CR, can influence the static and dynamic corporal position [32,33]. These corporal alterations can be observed when changing from a maximal occlusion to a centric occlusion due to a change in the muscle tone caused by an increase in the isotonic muscular tension in MI of the craniomandibular muscle complex, neck muscles, back and shoulders; while in RC the body achieves a more perpendicular position accomplishing a lower muscular work [33].

However, there is currently no consensus between researchers with regard to the primary cause of these alterations [34]. On this premise, the body should be treated holistically where health specialists work as an interdisciplinary team [35,36].

Knowing the different research, performed within the scope of Dentistry through the use of the optic reference system, can help us to three-dimensionally understand the manner of the mandibular motion itself, and that those of the adjacent structures as well as the stomatognathic system, can intervene in the body posture and in the locomotor system, allows the user to address problems comprehensively.

Taking into consideration these premises, an exhaustive bibliographic search was conducted in order to identify research publications in the scientific literature, in which the main theme is the gold standard motion capture system at the service of Dentistry.

## 2. Materials and Methods

The protocol of this review was designed according to the recommendations of PRISMA 2020 statement (Preferred Reporting Items for Systematic Reviews and Meta-Analysis), whose test results were reported according to the PRISMA 2020 guideline [37] (Appendix A). 

### 2.1. Searching Strategy

In order to search for scientific information, data sources such as Dentistry & Oral Sciences Source, Scopus, Web of Science, PubMed, CINAHL y SPORTDiscus were used in April 2022 by two independent investigators (B.M.-S., M.D.-P.). Exploration equations: “vicon” was used as a unique term for the basis of information in oral and dental science Dentistry & Oral Sciences Source; for the rest of databases the term “vicon” was used as well as orofacial nomenclature and synonyms of the same type, where the exploration equation used was: (dentistry OR stomatology OR “facial bones” OR “facial muscles” OR “mandible kinematics” OR “mandible dynamics” OR “oral function” OR “temporomandibular joint” OR “stomatognathic system” OR bite OR feeding OR jaw OR mandible OR lip OR mastication OR munch OR mouth OR oral OR tongue OR tooth) AND “vicon” (Table 1).

### 2.2. Selection Criteria

Scientific articles as well as doctoral theses, proceedings of scientific journals and books in any language that belong to the biomedicine field, from January 1984 to April 2022 were selected. The entire text from the article must be available either by public access or through the request for a library loan. Experimental research was also accepted, including observational and transversal studies, as well as case-control in human population, with all of them being based on the gold standard system, whether or not in combination with additional devices that were necessary to carry out the research (e.g., pressure sensors, force plate, electromyogram, video camera, microphone). Thematic studies outside of the odontological environment, clinical cases, bibliographical reviews or meta-analysis, grey literature as well as articles that had not resorted to the gold standard system as the main tool for human motion analysis were not included.

### 2.3. Selected Studies and Data Extraction

The search strategy consisted of the initial identification of Medical Heading Subjects (MeSH) and the Descriptors in Health Sciences, which were used together with the Boolean operators. 

The PICO strategy of research was selected, from which the following question was raised: Which are the most relevant articles published in Dentistry through the use of the gold standard human motion capture system between 1984 and 2022?

During the first phase, the key words described in Table 1 were searched in the different data banks. Duplicate articles were identified and successively the remaining articles were analysed through the reading of the title, summary and methodology aiming to determine if they fulfil the inclusion/exclusion criteria. Finally, the reading of the entire text to verify the eligibility was performed and included in the bibliographical review. 

All the articles selected by both investigators were collected through the bibliographic manager, Mendeley Desktop 1.19.8 (Mendeley Ltd., New York, NY, USA), where a shared folder was created in order to facilitate its classification and the choice of the selected literature review, as well as the structuring of the gathered data. Both investigators (B.M.-S., M.D.-P.) agreed with the choice of the majority of selected studies, discussing the eligibility of the articles in discordance for its final inclusion. 

The same procedure that was performed by both investigators was used to determine the presence of bias according to Joanna Briggs Institute (JBI) checklists. 

## 3. Results

A total of 186 articles were found in the scientific-technical databases through the searching criteria previously described. Subsequently, the PRISMA 2020 flowchart was used for their selection (Figure 1). A total of 23 articles (Table 2) that fulfilled the selection criteria were included in the systematic review.

### 3.1. Description of the Study Characteristics

#### 3.1.1. Geographic Origin

Most of the studies that were included in this bibliographical review were developed in Europe (34.78%) and North America (USA) (30.43%), followed by Asia (21.74%), Oceania (8.70%) and South America (4.35%).

#### 3.1.2. Temporality in Publications

From the 23 selected articles, the established range with regard to the dissemination of research in relation to the gold standard is 1992–2021. In the field of Dentistry, it has been over 15 years since the first publication was created for medical purposes, coming from Germany [38], the pioneering country in this sector. In almost 30 years, 22% of publications were released in 2015, with Germany [39,40,41] and United States [42,43] being the countries where these pieces of research were developed. From the initial publication, it took 10 years for other research studies about optical capture systems in 3D were made known. Subsequently, from 2003 to 2006, they have not been noticed in this respect. The countries where the research has been developed, apart from those already mentioned, are: France [44]; Taiwan [45,46,47]; New Zealand [48]; Brazil [49]; Australia [50]; Poland [51,52]; Japan [53,54]; and Spain [55].

#### 3.1.3. Funding

Among the included publications, five of them (27.74%) had private funding [48,53,54,56,57], one (4.34%) obtained mixed funding [50] and the remaining 17 (65.22%) did not achieve a grant [38,39,40,41,42,43,44,45,46,47,49,51,55,58,59,60]. 

**Table 2 sensors-22-05769-t002:** Summary of the studies included in the systematic review. The studies are listed in chronological order.

No.	Author (Year) Country	Study Design	Sample Size (*n*)	Study Variables	Most Relevant Results	Conclusions
1	Stüssi et al. (1992) Germany [38]	Observational Cross-sectional	*n* = 2 Adult with muscle transplant/healthy	Buccinator muscle kinematics: maximum contraction length, speed and fatigue (Vicon 3.5, Vicon Motion Sustems Ltd., Oxford, UK)	Healthy side’s buccinator muscles length during contraction was 6 mm, while the operated side was 3 mm. Muscles movement amplitude increases with time. Contraction operated patients were 2–3 times slower	The three-dimensional measurement of fixed points allows one to objectify surgical and rehabilitation results
2	Egret et al. (2002) France [44]	Observational Cross-sectional	*n* = 6 Adult golfers with/without mandibular repositioning orthopaedic appliance	Kinematic pattern in a golf swing: pelvic joint; shoulders; temporomandibular joint rotation angles; elbow joint flexion and ball speed changes at impact (Vicon)	The ball’s average speed is greater if the golfer wears the repositioner (133.02 km h^−1^ 13.88 vs. 125.04 km h^−1^ 14.83). There are significant differences with it (*p* < 0.03)	Mandibular repositioning has no effect on the golf swing kinematic pattern. However, if the golfer wears the splint at the moment of the impact, the ball speed is more regular
3	Hong et al. (2007) Taiwan [45]	Observational Cross-sectional	*n* = 20 Children with/without CP	Lip and jaw movement and speed during speech. (Vicon Motion 370)	The greatest discrepancy was observed in lip and jaw mobility in the group with CP during the pronunciation of monosyllables (*p* = 0.015) and polysyllables (*p* = 0.007). Children with CP showed a greater lip and jaw opening in pronunciation (*p* = 0.032); (*p* = 0.035); (*p* =0.024)	Children with CP presented high temporal oromotor variability and need to make a greater effort to coordinate the lip-mandibular movement on speech act
4	Röhrle et al. (2009) New Zealand [48]	Observational Cross-sectional	*n* = 1	Masticatory mandibular dynamics (Vicon MX)	The average values of mandibular trajectory during soft food chewing was 63.63 and 66.64 for hard foods	This method can be implemented in the fields of Dentistry and in the food industry biomechanics
5	Chen et al. (2010) Taiwan [46]	Observational Cross-sectional	*n* = 20 Children with/without CP	Speech motor kinematics: lip displacement, mouth opening’s length and speech (Vicon Motion 370)	The mean of the maximum oral opening of children with CP was 1.17 cm during the monosyllables’ pronunciation and 1.84 cm of polysyllables. The average speed of the maximum oral opening while pronouncing monosyllables was 42.4 and 73.5 cm/s while pronouncing polysyllables	Children with mild spastic cerebral palsy may have more difficulty with greater oromotor variability compared to healthy children
6	Hong et al. (2011) Taiwan [47]	Observational Cross-sectional	*n* = 24 Children with/without CP	Spatio-temporal index, maximum mandibular duration and speed and maximum lower lip displacement during speech (Vicon 370)	CP patients with spastic tetraplegia had higher speed rates when pronouncing monosyllables, no significant change with the lower lip’s duration and displacement	Speech motor control kinematic data provided enables clinicians to understand it in CP patients
7	Jarjura et al. (2012) Brazil [49]	Observational Cross-sectional	*n* = 12 Adults	Displacements between rest and maximum facial muscle contraction	Wrinkling forehead facial movements, frowning, smiling, and blinking are in a range between 11–13 mm of contraction	Normal patterns for facial muscle contraction were obtained despite a great individual variability
8	Moss et al. (2012) USA [60]	Observational Cross-sectional	*n* = 18 three children groups, one group without speech disorders and two groups with speech disorders	Mandibular coordination and labial dynamics during speech (Vicon 460)	Children with apraxia have fewer motion possibilities for an accurate speech	Some aspects of coordination can differentiate children without deficits from those with articulatory or phonological deficits
9	Ward et al. (2013) Australia [50]	Observational Cross-sectional	*n* = 12 Children with severe-moderate speech disability associated with CP/healthy	Mandibular and labial kinematics before, during and after participating in a motor speech intervention program (Vicon Motus 9.1)	All participants showed, in the jaw and lip, significant changes in motion features and significant changes in speech intelligibility in five of the six participants were associated	This study provides evidence supporting the use of a treatment approach aligned with a dynamic systems theory to improve motor speech movement patterns and intelligibility in children with CP
10	Maurer et al. (2015) Germany [39]	Observational Cross-sectional	*n* = 20 healthy adults	Symmetrical changes of movements on ankle, knee, hip, spine and neck in four different occlusal positions (Vicon MX T10)	The resting position was significantly more asymmetric than any of the three splinted positions (*p* = 0.049). There are no found differences between positions with splint	The use of splints increases the symmetry of the gait pattern
11	Hellmann et al. (2015) Germany [40]	Observational Cross-sectional	*n* = 12 healthy adults	Joints kinematics of ankle, knee and hip and lower extremities muscles electromyographic activity during one leg and bipedal posture in centric relation and maximum biting (Vicon MX)	There was no significance between the ankle, knee and hip joints, or mean angle values with the two mandibular positions. The kinematic values of the three studied joints revealed a significant reduction in both mandibular positions	Mandibular position only affects neuromuscular co-contraction patterns
12	Ringhof et al. (2015) Germany [41]	Observational Cross-sectional	*n* = 12 healthy adults	Trunk and head postural stability and kinematics in centric relation and submaximum biting (Vicon)	Maximum intercuspation significantly reduced the sway area on the postural platform	The physiological response to isometric activation of mastication muscles raises questions about the potential of oral motor activity as a strategy to reduce fall risk among patients with compromised postural control
13	Reuterskiöld and Grigos (2015) USA [42]	Observational Cross-sectional	*n* = 16 Kids distributed in two age ranges (children and adolescents)	Mandibular movement kinematics duration and variability during pronunciation (Vicon 460)	Mandibular duration and dynamics vary in different levels between children and adolescents depending on the length of the word	Young children show more durable mandibular movements than adolescents. It is possible that between the ages of six and fourteen skills that improve the repetition capacity are performed
14	Grigos et al. (2015) USA [43]	Observational Cross-sectional	*n* = 33 Kids distributed in three age ranges: Apraxia/speech delay/typical development	Mandibular and lip movements duration, speed, displacement and variability during word pronunciation (Vicon 460)	Significant differences in movement duration were observed in children with apraxia. Pronunciation variability of long words favours the variability of the motion variability differed between the group with delay and typical development	Kinematic differences between participants suggest a different response to the same linguistic challenges
15	Laird (2017) USA [59]	Observational Cross-sectional	*n* = 26 healthy adults	Occlusal morphology and mandibular dynamics during chewing of two types of food	The chewing cycle duration does not change significantly during chewing (0.65 s). Significance between the number of cycles and duration. Lateral displacement (8.32–9.25 mm)	The number of chewing cycles and bolus properties determine mandibular kinematics more than occlusal morphology. However, the last one determines the cycles and mandibular dynamics
16	Grigos et al. (2018) USA [56]	Observational Cross-sectional	*n* = 16 Children with/without apraxia	Facial cinematics after the pronunciation of the first syllable for each word (Vicon 460)	Unlike children without neurological disorders, those with apraxia showed a greater oral displacement and less variabilities of this movement	Changes in mandibular opening in patients with apraxia help the child to improve their pronunciation the words previously pronounced
17	Syczewska et al. (2018) Poland [51]	Observational Cross-sectional	*n* = 30 16 adults with fibula graft and 14 with iliac crest graft	Gait kinematics (Vicon 460 y MX)	The gait variables that statistically changed after surgery were: pelvic rotation, hip range motion in the sagittal plane, knee range motion in the sagittal plane, and operated side in a significant manner	The primary gait deviations that happen after surgery and the compensatory mechanisms that arise later, depend on the ubication of the graft donor site. Fibular graft patients have fewer gait problems than iliac crest grafts
18	Small et al. (2018) USA [58]	Observational Cross-sectional	*n* = 39 Four population groups based on age	Mandibular dynamics during protrusion and right-left lingual laterality. Age influence on the study variables (Vicon 460)	The range of values corresponding to mandibular movement oscillated between 0.69% and 24.23%. (mandibular movement not clinically observable until a clearly observable movement)	Age was not significantly correlated with the dynamics of the study
19	Nakamura et al. (2019) Japan [54]	Observational Cross-sectional	*n* = 20 healthy adults	Lip closing pressure and lip function during the intake of food.	The pressure during the intake of 10 g was significantly lower than during the intake of 3 g (*p* < 0.01)	Excess food causes food spillage
20	Kawaler et al. (2019) Poland [52]	Observational Cross-sectional	*n* = 5 healthy adults	Facial kinematics during English speech (Vicon 2018)	The shape and size of the facies slightly affect the quality of the reconstruction. The beard generates errors in the oral area	Easy muscle movements can be recorded using reflective markers in FMC (Face Motion Capture) technology. For greater language precision, it is recommended to increase the number of markers in the oral area
21	Kopera et al. (2019) USA [57]	Observational Cross-sectional	*n* = 24 three children groups, one group without speech disorders and two with disorders	Mandibular duration and dynamics during acoustic parameters	There were no significant differences between the groups with mandibular displacement, however between the group with apraxia and with normal development some significance was found with regards to motion duration after pronouncing certain syllables	The difficulty in controlling the duration of mandibular movement is a possible cause of lexical accentuation errors
22	García et al. (2020) Spain [55]	Observational Cross-sectional	*n* = 1	3D kinematics models	After studying the mandibular dynamics, models are built allowing mandibular advancement	Application: treatment of obstructive sleep apnoea
23	Sasakawa et al. (2021) Japan [53]	Observational Cross-sectional	*n* = 15 healthy children	Lip closing strength, lip dynamics and the movements spoon does (Vicon)	Lip seal strength varies depending on the type of food	It is necessary to consider the importance of food diversity and to pay attention to the spoon withdrawal period and lip function maturation

CP = cerebral palsy.

#### 3.1.4. Characteristics of the Sample

The sample size that was used in publications was not very large, being a total of 384 participants, with an average size sample of 16.69 participants and a median of 16. 

Most of the selected articles’ subjects were under-aged [39,42,43,45,46,47,50,53,56,57,60], and six publications included adults [39,40,48,51,52,54,59]. Only two research studies handled wide age ranges, basing their studies on children and adults [44,58], while four studies did not specify the sample [38,49,52,55]. The lowest sample size is one [48,55] and the highest 39 [58]. The average sample size on the series of studies is 16.69. The overall number of adults studied was 156, children/teenagers, being the majority of the population group with a total number of 228. The minimum age is three years old and the maximum is 70 years old (Figure 2). Of the total sample from all the studies registered, 293 were healthy individuals [38,39,40,41,42,44,45,46,47,48,49,50,52,54,55,56,57,58,59,60] whereas there were only 58 subjects with speech disorders [43,56,57,60], 38 subjects were patients with cerebral palsy [45,46,47,50], one had previously been submitted to be studied for facial surgery with a muscle transplant [38], and 30 presented facial bone reconstruction with fibula graft or iliac crest [51].

Only two studies with a basic theme of Dentistry excluded patients with abnormalities, traumas or maxillary surgeries during the last two years, and included those with orthodontic or orthopaedic treatment in progress and without mandibular blocks [39].

#### 3.1.5. Themes 

Different topics were studied, all of them were related to the stomatognathic system in a direct or indirect way, which remains the most studied topic (43.48%) the speech properties, especially pronunciation [42,43,45,46,47,50,52,56,57,60]. Dento-oro-facial level changes that influence posture and body movement [39,40,41,44,51] were also a reason for study (21.74%), as well as facial muscle analysis (17.39%) [38,49,53,54], mandibular dynamics during chewing (8.695%) [48,59] and mandibular kinematics in laterality and protrusion movements (8.695%) [55,58]. The selected research studies that were published in different journals and themes can be seen in Table 3.

#### 3.1.6. Design of the Research Studies

All the research studies published were classified within the observational and transversal category, eleven of which [38,43,44,45,46,47,50,51,57,58,60] will not be classified as they should be, despite the fact that they present case-control studies, due to the very short sample size that was used in their investigations.

#### 3.1.7. Duration

The duration of the studies ranged from one session [39,40,41,42,43,44,45,46,47,48,49,52,53,54,55,58,59,60] up to approximately 40 weeks [50], with an estimated average duration of four weeks and the median was one session, although there was one study that did not specify the number of sessions that were carried out [57]. 

#### 3.1.8. Relationship between Articles

The data that was used in one of the studies belonged to a previous research [56]; the authors of two publications shared their experimentation subjects in their studies [40,41] and another study [57] included a section of the sample that was used in other research [43]. Finally, three research studies aimed to continue previous scientific articles [55,56,57]. 

### 3.2. Area of Research

All the research studies included in the current article were selected according to structural involvement related to the stomatognathic system (Table 4). 

### 3.3. Marker Arrangements on Head and Facies

Different research studies suggest very similar locations in the category of stomatognathic system and in areas around them [38,39,40,41,42,43,44,45,46,47,48,49,50,51,52,53,54,55,56,57,58,59,60] Figure 3, Figure 4 and Figure 5.

### 3.4. Technical Features of the Gold Standard System 

The three dimensional motion capture systems used are: Vicon 3.5 [38], Vicon 370 [45,46,47], Vicon 460 [42,43,49,56,58,60], Vicon MX [40,48,51], Vicon 9.1 [50], Vicon MX-T10 [39]. The technical characteristics used in the series of studies are:Recording frequency of cameras of with a range of 60–200 Hz [38,39,40,41,46,53].Camera resolution of one megapixel.The VAX 3100 system to rebuild the three dimensional coordinates of the markers [44].Sampling rate 370 frames per second [48], 200 frames per second [59], 120 frames per second [42,43,52,56,58,60] up to 100 frames per second [49,54], 50 frames per second and 10 frames per second per every 0.083 s [57].Cameras T160 and storage of 16 MP [60], cameras in the infrared spectrum [52] and matrix of 2.1 MPX (1920 × 1080 px) [52] and high speed digital cameras of four interconnected megapixels [48].Kinematic data processed with MATLAB, version 7.5 [60], Software MATLAB, version 7.2 [42], software POLYGON [51] and BLADE [52].Body markers Plug-In Gait [40,41,51,57], markers OptiTrack [52], static markers with absolute error de <1 mm [38], reflective markers [39,40,42,43,45,46,48,56,58,59,60] and retro-reflective markers [49,50].

## 4. Discussion

This is the first systematic review performed over the application of the gold standard motion capture system in the Dentistry field. In the search for literature only 23 articles were found in relation with stomatognathic system, from which all of them were included in this literature review aiming to cope with as much existent information as possible over the use of referential optical system applied to dental sciences. 

Five articles [40,48,53,54,59] could be classified within pure dental sciences where authors studied occlusion [40], swallowing [53,54] and mastication [48,59]; five research studies relate the field of Dentistry to other sciences [39,41,44,55,58] such as Posturology [39,41], Sport Sciences [44], motor coordination [58] and Sleep Medicine [55]; the remaining investigations [38,42,43,45,46,47,49,50,51,52,56,57,60] analysed the different components related to the stomatognathic system in an indirect way. Research studies that focus on speech [42,43,45,46,47,50,52,56,57,60] have been selected not only as this is a part of the stomatognathic apparatus, but also to study the mandibular and labial dynamic.

### 4.1. Quality Evidence

For the critical assessment of the methodological quality in the included research studies in the present review and in order to determine the presence of bias, the Joanna Briggs Institute (JBI) [61] checklist was selected for observational cross-sectional studies. The JBI scale assigns a rating from zero to nine points to the selected articles. The majority of these publications obtained scores over six points, indicating an optimal methodological quality in their studies [40,41,43,44,45,46,47,50,51,52,53,56,57,58,60]; seven publications [39,42,48,49,54,55,59] obtained a middle level score due to the inaccuracy of inclusion criteria as well as the absence of a comparison group and the sample size; the worst score was obtained by one article [38], the score below four points being the only choice due to sample size, the participants, data collection and their analysis.

The sampling frame (First point of the JBI checklist) seemed dubious in all the articles due to the lack of an explanation and also due to its own contextualization. Additionally, almost all of the studies presented a short or deficient sample size (point number three in the JBI scale) where the number of participants was between one and 39. As a consequence, studies could be qualified as inappropriate due to the inability to detect significant differences in order to obtain conclusive results [62].

Twelve studies did not present a comparison group (fifth point of JBI checklist), which may increase the presence of bias causing the obtained results to be interpreted in a wrong way as a result of the absence of a control group [63].

### 4.2. System Application 

Vicon is still one of the main optoelectronic motion capture tridimensional systems based on markers [6], and traditionally has been used for the analysis of biomechanics, gait and robotics [64]. It is a precision system that analyses motion through video and it is controlled by a computer allowing the analysis of the dysfunctions of the human corporal dynamics [65]. The first research studies based on the use of the gold standard were focused on the clinical analysis of motion and human gait [66,67,68,69,70]. Over the years the application of this system has given rise to numerous investigations in the field of biomedicine, as well as the studies performed on Dentistry, research studies that link Dentistry with other sciences and studies that analyse elements of the stomatognathic system.

Knowledge of masticatory mandibular dynamics with the gold standard system is in service of prosthodontics, allowing implants design, crowns, and fixed partial dentures, in addition to restoring therapeutics in order to identify eccentric premature occlusal contacts during the mastication process [55], and this contributes to maintaining the coordination and health of dento-oro-facial system [71].

The method is useful for behavioural sciences, biomechanics and the food industry [48]. It can contribute to developing therapeutic protocols targeted to improve motor speech centre [50], and as a result the existing methods are not an ultimate solution in view of these types of problems [72].

Research has shown that using a repositioning splint during gait increases the symmetry of run pattern and it could help prevent injuries and improve performance [39]. Therefore, as certain authors show [33], the occlusion in maximum intercuspation and centric relation are related to functional body posture.

The occlusal plane inclination influences the kinematics of the masticatory cycle, limiting the jaw movement during closing [59]. Recently, particular attention has been attached to mandibular kinematics with the gold standard system in order to design mandibular advancement devices that allow greater opening ranges and thereby favouring the treatment of obstructive sleep apnoea [55]. Current devices are manufactured without taking into consideration anatomical features of the temporomandibular joint and without individualizing mandibular morphology of patients. As a consequence, in order to avoid a mandibular retrusion in certain areas where the upper respiratory tract could become narrowed, the individualization of the case is necessary as not all patients move their lower jaw equally [73].

### 4.3. System Management

#### 4.3.1. System Calibration

At the start of the study, and before initiating the record, it is recommended [39,48,49] to calibrate the capture system and for the participants to undergo a previous test. Ward et al. [50] reflect the importance of the responsible investigator to be qualified and trained from a methodological approach. After having informed the participants, the markers are properly placed, and cameras record the image from these markers while they are connected to a device, which synchronizes its functions and transmits these images to a particular computer and software. Some authors calibrate the position for each marker from a reference cube of 24 × 24 cm with eight spherical markers: head, forehand, nose, chin, lips (commissures, Cupid arc, middle and lower point) [50] and even 10 markers [42].

#### 4.3.2. Individual Placement

Researchers recommend that the subject should be in an upright position to achieve the most reliable calculation of the spatial coordination [38,46,48,53,54] or erect and with arms at rest [40], except for when the research focused on the study of the four main jaw positions (centric relation, centric occlusion, miocentric and maximum intercuspation) and its relationship with the gait. In this particular case, a 15 m walkway is installed, which is visualized by eight infrared cameras [39]. At times, when the body kinematics is analysed the body mass index of the participants is registered [40].

#### 4.3.3. Markers and Placement Selection

It is crucial to consider a proper selection of size [74] and marker locations in order for an optimal analysis of the structures to be conducted [75]. Other motion capture systems without markers have obtained mixed results in the location of the areas in the human body that are to be analysed, compared to the gold standard system having variations from 2 cm to 8 cm, increasing this discrepancy during motion [74,76].

The researchers encourage the installation of cameras around the capture space at the working room, aiming to avoid hidden markers. Although the gold standard system supplies hemispherical markers of 14.5 and 9.5 mm, they are not appropriate for light reflection in small areas such as the face and eyelids. As a consequence, spherical markers [54] have been created with a 4 mm diameter covered with reflective tape [49]. Some researchers also use reflective markers of 3 mm [42,43,56,58,60] in the upper lip, lower lip, corners of the mouth, the right side jaw, the left side jaw, and the middle jaw. The OptiTrack markers have semispherical shape and they are covered by a reflective sheet and are also covered by a cosmetic glue that allows easy application and removal [52]. Although for the recording of facial motions the use of small markers is recommended [76], the use of big markers increases the precision of the data registered [74].

The kinematic data are processed with a kind of software that does not need a facial model. Some authors considered strategic points of the face for 11 markers: central point of forehead close to the hairline, supraciliary, malar regions that coincide with eye canthus, pars centralis in both upper eyelids, labial commissure and chin [49]. The nasion marker and the one on the chin show the vertical movement and the head rotation [56,58].

Some other authors aiming to register the facial dynamics placed a total of 44 markers, 18 of which corresponded to the stomatognathic area. Subsequently, they corrected the number, and reduced it to 28, since this number is sufficient to register the facial dynamics [52]. In order to register lingual dynamics (protrusion and literalities), some researchers place eight reflective markers of 3 mm in the facies, three in the jaw (middle line and mental protuberance and on both sides of this) and five referential markers (middle line and on both sides of frontal) in the nasion and nasal appendix. In addition, one marker is placed on the tip of the tongue using denture adhesive [58].

With the objective of studying kinematics of the buccinator muscle, its length and contractibility, they are used when placed in the corners of the mouth and jugal areas and it is registered together with other facial points with four cameras stacked, one above another, on the same vertical axis of the right and left cheek, respectively, allowing the complete detection of the six markers, with two of them in the area outlined above [38]. 

By studying the masticatory kinematics and occlusal morphology, six reflective markers were secured to the skin using double sided adhesive tape, onto the surface, osteological referential points, pogonion, nasion, right and left lateral condylion, right and left gonion. According to this researcher, the position of the marker during the register, and also due to the skin displacement, could provoke a slight variation although it may only be a minor one [59].

One of the gold standard motion capture system models has been designed to automatically track the movements through contrast detection between a circular marker and surrounding pixels. For that purpose, an adjacent light source for each camera has been used and retro reflective markers (bright white against a darker environment) [50]. 

#### 4.3.4. Auxiliary Devices for the Markers’ Placement

Some researchers use 10 mm markers that are attached to the temple of a spectacle frame as a reference coordinate system [54] as well as taking measurements with miniature pressure sensors in order to register the labial seal [53], which are water resistant [54].

Some authors use markers with a 25.4 mm diameter fixed on the skin using double sided adhesive tape [44]. Among other points, the tragion point is registered so as to determine angulations at that level, while the participants wear a splint or replacement mandibular devices to determine if the mandibular position together with other joints are affected by the speed of the golf ball. In the current study none of participants show clinical or historical evidence of the temporomandibular joint dysfunction or the myofascial pain dysfunction syndrome or occlusal alteration [44].

The motor control of speech in a given population can be studied [45,46] through reflective markers with a 0.6 diameter [46] over an adjustable plastic facial mask with straps, which cover the upper half of patient: pars anterior head, preauricularis and nasal area. Registering labial dynamics and buccal opening speed with markers in commissures in the upper and lower lip [45,46,47]. The mandibular and lingual dynamics record is as important as recording the movement markers in the right, left and middle forehand, in the nasion and in the nose, as these are referential markers in order to take into account head rotation [43,60].

With regards to the analysis of masticatory trajectories, there is another alternative, which consists of designing a special device for monitoring the mandibular movement. Eight cameras arranged in a semicircle record six reflective markers, three of them are placed at the nasal bridge and two of them on the forehand [48]. Due to the fact that the skin has minimal movement during mastication, the rest are arranged over the dental device that has been designed. The participant must be in occlusion to be able to chew food with different textures later [48].

In addition, it is registered together with kinematic corporal mandibular dynamics, and the researchers use up to 13 cameras and whole- body markers of Plug-In-Gait, also registering electromyographic activity of the rectus femoris muscle, medial vastus muscle, biceps femoris, tibialis anterior, the soleus muscle, the medial gastrocnemius and the masseter [40]. Two mandibular positions were registered; the centric relation and the maximal intercuspation [40]. If in addition to the kinematics of the trunk and head, the postural stability is analysed, some researchers are supported with the use of a force platform [41]. 

#### 4.3.5. Master Record

The marker mapping is determined by the kinematics features of the tissue to be studied [52]. Due to the great complexity of the stomatognathic system, the ideal mapping in virtual reality is quite difficult, as a result the temporary loss in the visibility fields of the camera, or by the excessive proximity of the neighbour markers, resulting in temporary data loss [52]. Some authors maintain that anatomical features, capacity for expression and intense facial hair can interfere in standard mapping of the markers [52]. Occasionally, it is recommended to adhere the markers to the skin through the use of a degreasing substance [49]. 

Binh et al. [77] suggest a mapping of markers for the study of facial and mandibular dynamics. In order to obtain varied and relevant information in terms of investigation, although there are templates in the literature in marker distribution [77], the authors call for the creation of a master record, from which markers are tested and defined to finally decide the number and location for further research. It is important to emphasize that a greater number of markers allows the improvement of the nuances of imitation, but it also increases the probability of having mistakes or fading.

### 4.4. Limitations 

Limitations exist in the present systematic review. Despite the fact that there is a bias to a large extent in some of the studies that were included due to the scarcity of the literature published in the Dentistry field, all the articles found by the researchers were included. Potential for bias was evaluated through methodological quality assessment of the articles through JBI protocol by means of its critical assessment guide.

Several problems were encountered in the methodological design in the majority of the studies, such as the contextualization of the sampling frame, as well as some deficiencies with regards to the determination of sample power being observed. There is only an article that talks of the need of training of researchers in using the optical system in question [50] and a minority of the calibration of the said technology prior to it being used [48,49,50,52]. In 2010, talks of the concordance intra-radar and inter-radar [46] and subsequently the concordance was mentioned in only one study [59]. As a result we assume that from a methodological perspective, when drafting the information regarding concordances and training, some of the studies have ignored or obviated these aspects [38,39,40,41,42,43,44,45,46,47,48,49,50,51,52,53,54,55,56,57,58,60]. 

### 4.5. Future Research

As a result of the scarcity of studies found, it is suggested one considers future studies through the gold standard system in the field of pure dental sciences. Therefore, such research can provide relevant data for better understanding of kinetics and mandibular kinematics in isolation and its relation to the kinematics of other body systems. On the question of the notorious influence between the static and dynamic relationship between the maxillary and lower jaw with regard to the rest of body structures [78], however, at the present time there is insufficient data. As a result, we believe that this optical capture system could encourage development in this field since the scientific literature on the matter at this point is insufficient.

## 5. Conclusions

This review identifies 23 articles that addressed issues related to Dentistry in a direct or indirect way. The studied topics were:the study of the speech properties (43.48%)dento-oro-facial level changes that influence posture and body movement (21.74%)facial muscle analysis (17.39%)mandibular dynamics during chewing (8.695%)mandibular kinematics in laterality and protrusion movements (8.695%)

Five research studies were found that focused exclusively on Dental Sciences, five research studies related the field of Dentistry to other sciences, and the remaining research studies analysed the elements related to the stomatognathic system in an indirect way. From a physiological and pathological perspective, the applications of the optic system in Dentistry are especially relevant. 

The markers’ arrangement depends on the dynamic to be analysed, and that is the great variability in anatomical references.

It can be concluded that the scarcity of literature on the gold standard motion capture system in the field of Dentistry, and the methodological errors that have been found concerning some research studies, shows the need for future investigative studies to be conducted in this area with this kind of advanced technology. 

## Figures and Tables

**Figure 1 sensors-22-05769-f001:**
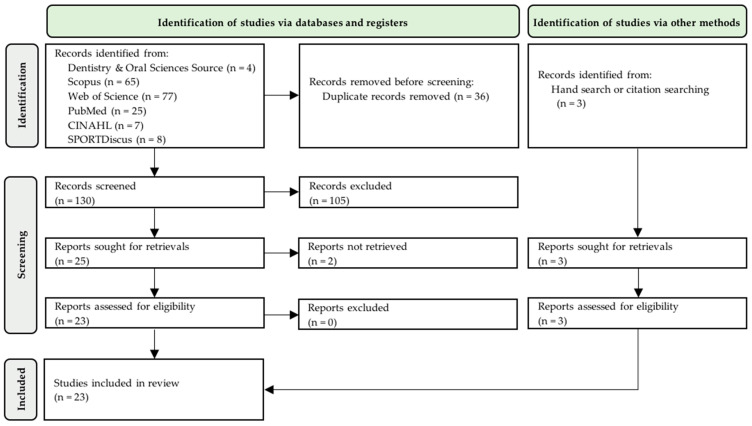
PRISMA 2020 flow diagram of literature search and selection process.

**Figure 2 sensors-22-05769-f002:**
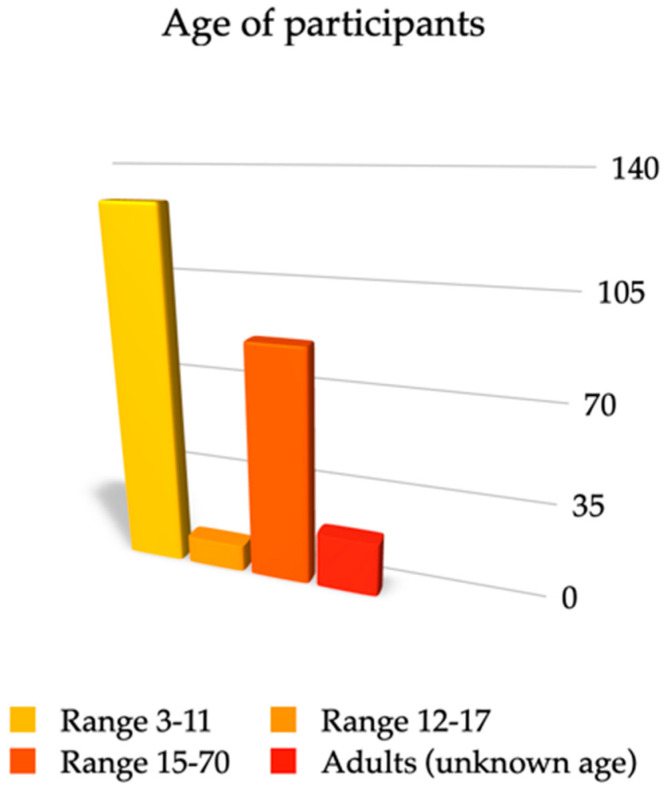
Age range of the population study.

**Figure 3 sensors-22-05769-f003:**
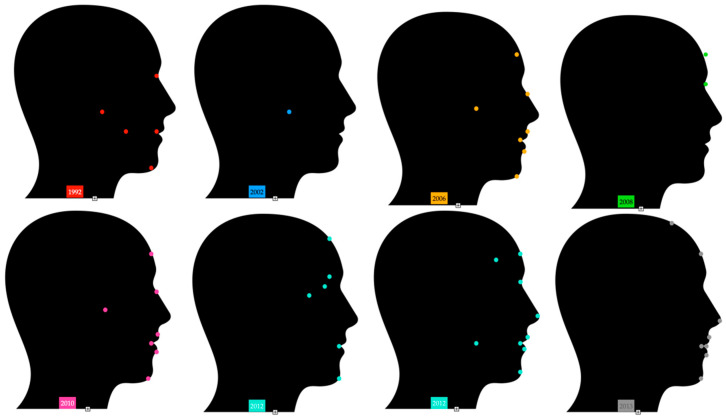
Marker locations in research studies from 1992 until 2013 [38,44,45,46,47,48,49,50,60].

**Figure 4 sensors-22-05769-f004:**
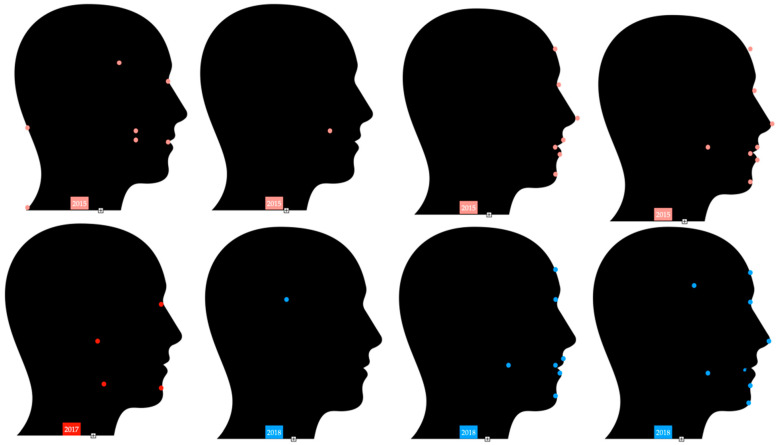
Marker locations in research studies from 2015 until 2018 [39,40,41,42,43,51,56,58,59].

**Figure 5 sensors-22-05769-f005:**
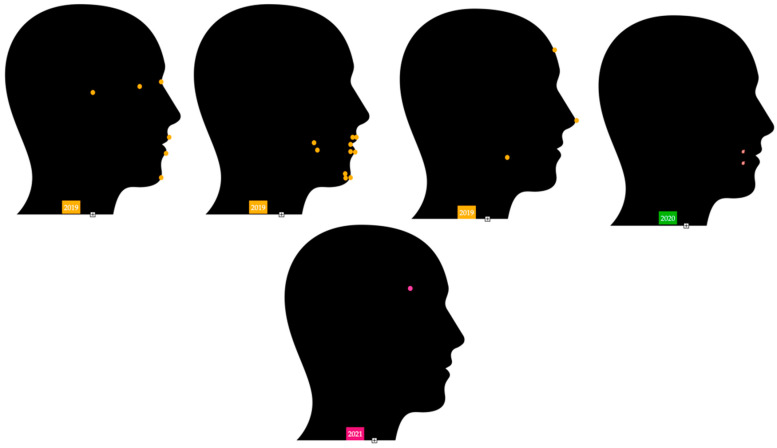
Marker locations in research studies from 2019 until present time [52,53,54,55,57].

**Table 1 sensors-22-05769-t001:** Searching equations between years 1984–2022.

Searching Equations	Database	Results
“vicon”	Dentistry & Oral Sciences Source	4
(Dentistry OR stomatology OR “facial bones” OR “facial muscles” OR “mandible kinematics” OR “mandible dynamics” OR “oral function” OR “temporomandibular joint” OR “stomatognathic system” OR bite OR feeding OR jaw OR mandible OR lip OR mastication OR munch OR mouth OR oral OR tongue OR tooth) AND “vicon”	Scopus	65
	Web of Science	77
	PubMed	25
	CINAHL	7
	SPORTDiscus	8

**Table 3 sensors-22-05769-t003:** Theme for the scientific dissemination and research journal.

Articles	Journal’s Names	Themes (General Issues Addressed in Journals)
Stüssi et al. (1992) [38]	Biomedizinische Technik	Biomedical engineering, medical informatics and biotechnology
Egret et al. (2002) [44]	International Journal of Sports Medicine	Training, orthopaedics, nutrition, behaviour, physiology, biochemistry, immunology and biomechanics
Hong et al. (2007) [45]	IFMBE proceedings	Biomedical engineering y bioengineering
Röhrle et al. (2009) [48]	Journal of Prosthodontics	Prosthodontics, implantology, aesthetics and restorative dentistry
Chen et al. (2010) [46]	Journal of Neuro Engineering and Rehabilitation	Neuroscience, biomedical engineering and rehabilitation
Hong et al. (2011) [47]	Research in Developmental Disabilities	Interdisciplinary and in direct relation to the understanding or the solutions to problems associated to developmental disabilities
Jarjura et al. (2012) [49]	Brazilian Journal of Otorhinolaryngology	Otolaryngology and associated areas (cranial-maxillofacial and phoniatrics)
Moss et al. (2012) [60]	Journal of Medical Speech-Language Pathology	Otolaryngology and rehabilitation
Ward et al. (2013) [50] Kopera et al. (2019) [57]	International Journal of Speech-Language Pathology	Otolaryngology and rehabilitation
Maurer et al. (2015) [39]	PLoS ONE	Natural science, medicine, social science and humanities
Hellmann et al. (2015) [40]	Human Movement Science	Human motion from the perspective of psychology, biomechanics and neurophysiology
Ringhof et al. (2015) [41] Sasakawa et al. (2021) [53]	Journal of Oral Rehabilitation	Rehabilitation and oral physiology applied
Reuterskiöld and Grigos (2015) [42]	BioMed Research International	Broad thematic journal, life sciences and medicine
Laird (2017) [59]	American Journal of Physical Anthropology	Physical and social anthropology
Grigos et al. (2018) [56]	Clinical Linguistics and Phonetics	Linguistics and phonetics of speech and language disorders
Syczewska et al. (2018) [51]	Acta of Bioengineering and Biomechanics:	Technique and medicine
Small et al. (2018) [58]	American Journal of Speech-Language Pathology	Speech and language pathology, hearing rehabilitation, augmentative and alternative communication, cognitive impairment, craniofacial disorders, swallowing and feeding
Nakamura et al. (2019) [54]	Physiology and Behaviour	Causal physiological mechanisms of behaviour and its modulation by environmental factors
Kawaler et al. (2019) [52]	Journal of Intelligent Information Systems	The integration of artificial intelligence and database technologies in order to create next-generation information systems
García et al. (2020) [55]	Bio-Design and Manufacturing	Mechanic engineering, the mechatronic devices and biomedical engineering

**Table 4 sensors-22-05769-t004:** Areas of research of the included studies and facial landmarks that have been used.

Article	Objective of Research Study	Facial Landmarks Used in the Study for Markers Placement
Stüssi et al. (1992) [38]	To assess the results achieved after a dynamic reconstruction of paralysis of the facial nerve, measuring facial distances and the length of buccinator muscles during contraction, also reflecting its speed and fatigue	Glabella point Tragion point (right and left) Labial commissure of mouth
Egret et al. (2002) [44]	To study the kinematic pattern of the golf swing, and the analysis of the temporomandibular joint, using jaw repositioning devices in order to cover occlusal surfaces and to increase 3 mm the vertical dimension	Tragion Dermographic projection of condyle
Hong et al. (2007) (2011) [45,47] Chen et al. (2010) [46]	To study the kinematic motor control in speech, registering at the same time the facial, mandibular and labial dynamic	Labial commissures Middle upper lineal labial Middle lower lineal labial
Röhrle et al. (2009) [48]	To generate a geometrical model of the dentition from a trajectory register of mastication and the construction of a device with the objective of registering such kinematics	Bridge of the nose Forehead Interincisal point
Jarjura et al. (2012) [49]	To study facial, labial and mandibular dynamics referring above all to muscle contraction in these areas	Forehead centre Central position above both the eyebrows Malar regions in the line of the eyes’ external corners Centre of both upper eyelids Corners of the labial commissures Chin
Moss et al. (2012) [60]	To conduct research on the coordination between mandibular and labial dynamics during speech	Upper lip Lower lip Corners of mouth Right jaw Left jaw Middle jaw. Right, left, and middle forehead Nasion Nose
Ward el al. (2013) [50]	To collect labial and mandibular kinematic data generated during the speech process	Forehead Mental protuberance of the chin Right and left corners of mouth Right and left upper points of Cupid’ s bow Mid-point located on the lower lip vermillion Upper lip vermillion
Maurer et al. (2015) [39]	To study the relationship between different mandibular positions and the gait pattern in four different conditions of occlusion	Neck Head
Hellmann et al. (2015) [40]	To determine the kinematics in several joints of human anatomy, among them the ankle, knee and hip joint, the electromyographic activity of the muscles of the lower limbs during bipedal stance and on one leg in centric relation and maximum intercuspation	Left front head Left back head Right front head Right back head
Ringhof et al. (2015) [41]	To study the kinematics of the trunk and head in centric relation and maximum intercuspation but they analyse postural stability	Left front head Left back head Right front head Right back head
Reuterskiöld et al. (2015) [42]	To analyse the mandibular dynamic during pronunciation	Midline of the vermilion border of the upper lip Midline of the vermilion border of the lower lip Superior to the mental protuberance of the mandible Corners of the mouth Nose, Nasion Forehead
Grigos et al. (2015) (2018) [43,56]	To study facial kinematics followed after phoneme articulation	Nose Nasion Forehead Midline of the vermilion border of the upper lip Midline of the vermilion border of the lower lip Corners of the mouth
Laird et al. (2017) [59]	To analyse masticatory kinematics and occlusal morphology	Condylion right and left Nasion Pogonion Gonion right and left
Syczewska et al. (2018) [51]	To study the gait kinematics after the reconstruction of facial bones with fibula graft or iliac crest	No facial landmarks
Small et al. (2018) [58]	To study the mandibular kinematics during protrusion and right and left lingual laterality, as well as the influx of age over the study variables, as it may affect the speech	Midline/mental protuberance and to the right and left sides Forehead at midline and the right and left sides Nasion Tip of the nose Tongue tip
Nakamura et al. (2019) [54] Sasakawa et al. (2021) [53]	To study the labial function and pressure during closing in food reception with and without a spoon	Superior lip Inferior lip Chin Right and left rims Right and left temples
Kawaler et al. (2019) [52]	To study facial kinematics followed after phoneme articulation	Hat Forehead Eyebrow Left and right eyes Left and right cheek Nose Mouth Chin
Kopera et al. (2019) [57]	To study mandibular duration and dynamics together with acoustic parameters	Midline of the vermilion border of the upper lip Midline of the vermilion border of the lower lip Right and left corners of the mouth and Jaw midline, right and left sides Forehead midline, right and left sides Nasion Nose tip
García el al. (2020) [55]	To study the mandibular kinematics in order to be able to subsequently design devices that allow greater opening ranges as treatment for obstructive sleep apnoea	Right and left condyle Lower incisor

## Data Availability

Not applicable.

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
