# Peer review of "(untitled)"

_sensors, 2022, doi:10.3390/s22155769_

Round 1
Reviewer 1 Report
Review,
Review of the article entitled Dentistry in combination with the gold standard motion capture system: systematic review.
This review is somewhat interesting in terms of evaluating the efficiency of the mandibular kinematics recording system, the authors researched the literature and selected materials dedicated to this topic. With the movement recording system one can obtain fast, accurate and non-invasive quantitative analysis of patients ’biomechanical and neuromuscular parameters, at the mandible level.
I have a few remarks to make about this article.
TITLE
1. The title does not seem appropriate. I suggest improving it. An example, it would be
REVIEW ON MANDIBULAR MUSCLE KINEMATICS
ABTRACT
2. I think it needs to be rewritten. It is very evasive. The reader does not understand the topic approached and the result of this approach.
INTRODUCTION
The introduction must be improved.
3. In order to approach the topic specified in the title, in the introduction it is necessary to individualize the mandibular kinematics, how it can be registered and what is the benefit of its registration. The purpose of this study is to highlight how to record mandibular movements, but the authors mix various analyzed movements as are the studies they have selected. I don't know if they can get something out of this review, because only 5 articles addressed aspects of dentistry.
4. The introduction is not structured.
5. The information is not related.
6. The authors begin the introduction with the Vicon® motion recorder (Oxford Metrics®, Oxford, UK) line and its applications in medicine in general and in other specializations. The authors refer only in a paragraph line 77-line 82 about the recording of mandibular movements. I consider that the mandibular movements, individually, related to the posture relation, the centric relation, and the maximum intercuspidation relation must be treated. These aspects are especially important because in every situation the muscle contraction is different. Moreover, addressing these issues briefly will guide the authors to organize the results and discussions.
MATERIALS AND METHODS
7. In this section, table 1 can be reorganized because they are the same search words. They do not have to be entered for each row in the table.
RESULTS
The material and method section needs to be improved.
8. Figure 2 is nice, but not very well understood and does not go too well in this scientific review. I suggest you remove it.
9. I suggest changing the term magazine to journal.
10. I suggest organizing section 3.3 Theme for the magazines of scientific dissemination and research line 188 with a table. It is not well highlighted in the text.
11. The same goes for subchapter 3.4 Area of research. Please organize the information. They are very different.
12. Please organize the section for clarity in transmitting the information
DISCUSSION
13. It is necessary to reorganize the discussions after reorganizing the results.
CONCLUSSION
14. The conclusions must be rewritten according to the results obtained.
BIBLIOGRAPHY.
The bibliography is written correctly and gathers a significant number of bibliographic sources.
Final remark.
Reconsider after the requested changes.
Author Response
Dear Reviewer:
Both authors appreciate your comments and suggestions, these have helped improve the way we present our research. Below we will detail and resolve each of your comments.

Reviewer 2 Report
1.
Authors should investigate book papers that are not published on the internet.
2.
Authors searched some websites for review. But, the discussion for review is short Reviewer cannot acquire knowledge on the gold standard motion capture system from this manuscript.
3.
Table2 list a summary of the studies included in the systematic review. But, there are few Asian researchers. Authors should add a new discussion about this reason.
Author Response

(The authors gave the same response as above.)

Round 2
Reviewer 1 Report
Reviews,
Thank you for the answers made in order to improve the results of this manuscript.
Although the authors have made substantial changes, I consider the following:
1. More effort is needed in reorganizing the tables.
2. The obtained results should be discussed more in the discussion chapter.
Author Response

(The authors gave the same response as above.)
